# ACE Inhibitory and Antihypertensive Activities of Wine Lees and Relationship among Bioactivity and Phenolic Profile

**DOI:** 10.3390/nu13020679

**Published:** 2021-02-20

**Authors:** Raúl López-Fernández-Sobrino, Jorge R. Soliz-Rueda, Maria Margalef, Anna Arola-Arnal, Manuel Suárez, Francisca I. Bravo, Begoña Muguerza

**Affiliations:** Nutrigenomics Research Group, Department of Biochemistry and Biotechnology, Universitat Rovira i Virgili, 43007 Tarragona, Spain; raul.lopez@urv.cat (R.L.-F.-S.); jorgericardo.soliz@urv.cat (J.R.S.-R.); maria.margalef.jornet@gmail.com (M.M.); anna.arola@urv.cat (A.A.-A.); manuel.suarez@urv.cat (M.S.); begona.muguerza@urv.cat (B.M.)

**Keywords:** blood pressure, Cabernet grape variety, hypertension, polyphenols, SHR, winery by-product

## Abstract

Wine lees (WL) are by-products generated in the winemaking process. The aim of this study was to investigate the angiotensin-converting enzyme inhibitory (ACEi) activity, and the blood pressure (BP) lowering effect of WL from individual grape varieties. The relationship among their activities and phenolic profiles was also studied. Three WL, from Cabernet, Mazuela, and Garnacha grape varieties, were firstly selected based on their ACEi properties. Their phenolic profiles were fully characterized by UHPLC-ESI-Q-TOF-MS. Then, their potential antihypertensive effects were evaluated in spontaneously hypertensive rats (SHR). BP was recorded before and after their oral administrations (2, 4, 6, 8, 24, and 48 h) at a dose of 5 mL/kg bw. Cabernet WL (CWL) exhibited a potent antihypertensive activity, similar to that obtained with the drug Captopril. This BP-lowering effect was related to the high amount of anthocyanins and flavanols present in these lees. In addition, a potential hypotensive effect of CWL was discarded in normotensive Wistar–Kyoto rats. Finally, the ACEi and antihypertensive activities of CWL coming from a different harvest were confirmed. Our results suggest the potential of CWL for controlling arterial BP, opening the door to commercial use within the wine industry.

## 1. Introduction

Nowadays, the leading cause of death worldwide is cardiovascular disease (CVD), being hypertension (HTN) one of their major risk factors [1]. The global prevalence of HTN is high since it is suffered by one in four adults [2]. In fact, a 25% reduction of its prevalence is one of the global targets to be attained by 2025 [3]. The adoption of healthy lifestyles in combination with pharmacological therapy has been shown to be effective for controlling blood pressure (BP) and improving CVD [4]. In this sense, angiotensin-converting enzyme (ACE) inhibitors, such as Captopril or Enalapril, are the first-choice treatments for HTN [5]. These drugs act blocking the ACE, which plays an important role in the BP regulation within the renin-angiotensin system [6]. In fact, its inhibition exerts a clear BP-lowering effect since ACE catalyzes the hydrolysis of the peptide angiotensin I (Ang I) to generate the vasoconstrictor Ang II. ACE is also involved in the kallikrein–kinin system, degrading the vasodilator bradykinin. Despite the effectiveness in controlling the BP of ACE inhibitors, new natural compounds are being investigated since drugs can cause certain side effects in some patients [7]. These alternatives could result in the reduction of HTN at the early or mid-stages of the disease [8]. Thus, antihypertensive compounds from natural sources have emerged as an excellent alternative to synthetic drugs, and are highly demanded, and researched.

Agri-food by-products have emerged as a novel source to obtain these natural antihypertensive agents since they can contain compounds with a wide range of biological properties [9]. The use of these waste products as a source of bioactive compounds allows their revaluation, making the food and agricultural industries more sustainable and environmentally friendly [10,11]. Grapes are one of the world’s largest fruit crops and the wine production process generates large amounts of by-products [12]. Some winery by-products, such as grape seeds and skin, have already been used as a source for the extraction of phenolic compounds with numerous health benefits, including antihypertensive properties [13,14,15,16,17]. However, the presence of antihypertensive compounds in other winery by-products as wine lees (WL) remains unexplored. 

According to the Council Regulation (EEC) No. 337/79, WL are “the residue that forms at the bottom of recipients containing wine, after fermentation, during storage or after authorized treatments, as well as the residue obtained following the filtration or centrifugation of this product” [18]. The potential application in the food, cosmetics, and pharmaceutical industries of WL has been suggested [19]. In fact, some studies have reported antioxidant, antimicrobial, anti-inflammatory, and cardioprotective properties of WL [19,20,21]. Differently from other winemaking by-products, WL have been least studied and exploited, and only the extraction of ethanol and tartaric acid is performed on a large scale [22]. However, WL could present ACEi and/or antihypertensive properties since their use as a source of phenolic compounds has been suggested [20]. In this sense, some studies have detected anthocyanins, flavonols, flavanols, and phenolic acids in WL [19,23,24,25,26].

Therefore, the aim of this study was to investigate the potential antihypertensive effect of WL. Thus, the ACEi activity was evaluated in five different WL generated from the winemaking process using single grape varieties. Three of these WL were selected according to their ACEi activity. Their phenolic profile was fully characterized and their antihypertensive activities were tested in spontaneously hypertensive rats (SHR). In addition, we evaluated BP-lowering effect of the selected WL in normotensive rats Wistar–Kyoto (WKY) to rule out a potential hypotensive effect. Furthermore, their ACEi and antihypertensive activities were evaluated using WL from a different harvest.

## 2. Materials and Methods

### 2.1. Chemicals and Reagents

Human Angiotensin-converting enzyme (ACE, EC 3.4.15.1, 5.1 U/mg), Captopril (Pub-Chem CID: 44093) and N-Hippuryl-His-Leu (Hip-His-Leu), were purchased from Santa Cruz Biotechnology (Dallas, TX, USA). O-aminobenzoylglicil-p-nitrofenilalanilprolina (o-Abz-Gly-p-Phe(NO2)-Pro-OH, PubChem CID: 128860) was provided by Bachem Feinchemikalien (Bubendorf, Switzerland). Acetonitrile and trifluoroacetic acid HPLC grade were purchased from Sigma-Aldrich (Madrid, Spain). Gallic acid, (−)-epicatechin, p-coumaric acid, and (+)-catechin were purchased from Fluka/Sigma-Aldrich; chlorogenic acid, caffeic acid, malvidin-3-O-glucoside, (−)-epigallocatechin gallate, and procyanidin dimer B2 were purchased from Extrasynthése (Lyon, France); cyanidin-3-O-rutinoside was purchased from PhytoLab (Vestenbergsgreuth, Germany); resveratrol was purchased from Carl Roth (Karlsruhe, Germany); and rutin was kindly provided by Nutrafur S.A. (Murcia, Spain). All other chemical solvents used were of analytical grade.

### 2.2. Wine Lees

WL were provided by Grandes Vinos y Viñedos, S.A located in the Cariñena P.O.D area (Zaragoza, Spain). They were collected after racking the wines. All the wines were elaborated with a single grape variety and following the same manufacturing procedure. The selected grape varieties were Cabernet, Garnacha, Mazuela, Merlot (all of them red grape varieties), and Macabeo (white grape variety). Moreover, lees obtained in the elaboration of wine with Cabernet grapes were supplied from two different harvests (CWL and CWL2). WL were centrifuged at 3000 × g for 15 min at 4 °C to remove solid particles. Supernatants were collected and kept at 4 °C until their analysis or administration to animals. 

### 2.3. Measurement of the ACEi Activity

ACEi activity was measured by a fluorescence technique according to Mas-Capdevila et al. [27]. This technique is based on the ability of ACE to hydrolyze the fluorescence compound o-Abz-Gly-p-Phe(NO_2_)-Pro-OH. Inhibition of this enzyme produces a decrease in fluorescence values. Thus, an aliquot of 40 μL of WL was added to a microtiter-plate well and mixed with 160 μL of 0.45 mM o-Abz-Gly-p-Phe(NO_2_)-Pro-OH dissolved in 150 mM Tris-base buffer (pH 8.3), containing 1.125 M NaCl. The enzymatic reaction started by adding 40 µL of an ACE solution prepared in 0.15 M Tris buffer (pH 8.3) containing 0.1 μM of ZnCl_2_ (enzyme concentration in the well was 0.04 U/mL). The reaction was carried out at 37 °C during 30 min. At this time point, fluorescence measurements using λex 360 nm and λem 400 nm were recorded and used to determine the inhibitory activity. The ACEi activity was calculated using the following formula: $$\mathrm{ACEi}\;\mathrm{activity}\;(\%):1-\frac{S-\mathrm{Bs}}{\mathrm{Pc}-B}\times 100$$
where S is the fluorescence emitted after the action of ACE on the substrate, with inhibitor (sample), Bs is the fluoresce emitted by the substrate and the sample, Pc is the fluoresce emitted after the action of ACE on the substrate, without inhibitor, and B is the fluoresce emitted by the substrate and the sample. 

ACEi activity was expressed as a percentage (%) or IC_50_ (µL). Percentage of ACEi activity was determined at WL volume of 0.16 µL in order to compare the effects of different WL on ACE activity. IC_50_ was calculated by linear approximation regression. Data are represented as the mean value of three determinations ± SD. 

### 2.4. Detection and Quantification of the Phenolic Compounds from Wine Lees

The individual phenolic profile of WL from Garnacha, Cabernet, and Mazuela WL was carried out by high-performance liquid chromatography coupled to electrospray ionisation and quadrupole time-of-flight mass spectrometry (UHPLC-ESI-Q-TOF-MS). WL samples were diluted twice with water:methanol with 1% of formic acid (50:50, v:v), centrifuged for 5 min at 17,150 × g at room temperature and supernatants were directly analyzed using a 1290 UHPLC Infinity II series coupled to a Q-TOF/MS 6550 (Agilent Technologies, Palo Alto, CA, USA). Two different methodologies based on UHPLC-ESI-Q-TOF-MS systems were used to separate, detect, and quantify the non-anthocyanin and anthocyanin phenolic compounds. For the separation of non-anthocyanin compounds, an Acquity HSST3 C18 column (150 mm × 2.1 mm i.d., 1.8 µm particle size) (Waters, Milford, MA, USA) was used and the mobile phase consisted of (A) water:acetic acid (95:5, v:v) and (B) acetonitrile. The gradient mode was as follows: initial conditions, 0% B; 0–0.5 min, 0% B; 0.5–18 min, 0–30% B; 18–21 min, 30–95% B; 21–24 min, 95% B; and 24–25 min, 100–0% B. A post-run of 6 min was required for column re-equilibration. The flow rate was set at 0.550 mL/min and column temperature was 45 ºC. The injection volume was 2.5 µL for all runs. Electrospray ionization (ESI) operating in negative mode was conducted with a gas temperature at 200 °C and the flow rate was 14 L/min. Nebulizer gas pressure was 20 psi, sheath gas temperature was 350 ºC, sheath gas flow was 11 L/min, and the capillary voltage was 3000 V. The anthocyanins compounds were separated on an Acquity BEH C18 column (100 mm × 2.1 mm, 1.7 µm particle size) (Waters) and the mobile phase consisted on water:formic acid (9:1, v:v) (A) and acetonitrile (B). The gradient mode was as follows: initial conditions, 0% B; 0–0.5 min, 0% B; 0.5–5 min, 0–9% B; 5–7 min, 9–15% B; 7–9.5 min, 15–30% B; 9.5–10 min, 30–100% B; 10–12 min, 100% B; and 12–12.1 min, 100–0% B. A post-run of 5 min was required for column re-equilibration. The flow rate was set at 0.4 mL/min and column temperature was 25 ºC. The injection volume was 2.5 µL for all runs. ESI operating in positive mode was conducted with a gas temperature set at 200 °C and the flow rate was 14 L/min. Nebulizer gas pressure was 20 psi, sheath gas temperature was 350 ºC, sheath gas flow was 11 L/min and the capillary voltage was 3000 V. The mass spectra were recorded between 100–1000 m/z at 2.5 spectra/s for both methodologies.

The assignment of the phenolic compounds was performed by direct comparison with the commercial standards available or by bibliographic information using chromatographic behavior, mass accurate molecular ion ([M-H]- or [M-H]+), and fragmentation patterns [19,28,29]. The obtained calibration curves of commercial standards available were used for the quantification of their corresponding phenolic compounds. When commercial standards were not available, a tentative quantification was carried out by using the calibration curve of the standard more similar. 

### 2.5. Experimental Procedure in Rats

Male SHR and WKY rats (17–20-week-old, weighing 310–350 g) were purchased from Charles River Laboratories España S.A. (Barcelona, Spain). The animals were housed at a temperature of 23 °C with 12/12 h light/dark cycles and 50% of humidity. After quarantine and a training period of 2 weeks, animals were given tap water and a standard diet (A04 Panlab, Barcelona, Spain) ad libitum during the experiments. The initial values of the systolic blood pressure (SBP) and diastolic blood pressure (DBP) in the SHR were 186.6 ± 1.7 and 153.6 ± 2.6 mmHg, respectively.

Figure 1 shows a graphical representation of the three experimental designs used in this study. A first study was carried out in SHR in order to evaluate antihypertensive effect of three WL obtained in the winemaking process with three different grapes varieties: Cabernet, Garnacha, and Mazuela (Figure 1A). For that, a single dose of 5 mL/kg bw of the WL was administered to SHR rats. Water and Captopril (50 mg/kg bw, dissolved in water) were used as a negative and positive control, respectively. A second study was carried out in WKY rats to discard a possible hypotensive effect of the Cabernet WL (CWL, Figure 1B). These WL were administered to animals in a single dose (5 mL/kg bw). Water was used as a negative control. In both studies, SBP and DBP were recorded in the animals before and 2, 4, 6, 8, 24, and 48 h after treatment administration to rats using the tail-cuff method, according to Quiñones et al. [30]. ∆SBP and ∆DBP were calculated as the difference between the mean values of SBP or DBP after and before treatment administration for each rat. Data were expressed as the mean values ± SEM for a minimum of six experiments.

For the evaluation of the effect of different harvests of CWL on the decrease in BP, an additional trial was conducted with SHR (Figure 1C). The study was carried out by administering a dose of 5 mL/kg bw of CWL2 to SHR (*n* = 6 per group). Water was used as a negative control. BP was recorded before and 6 h after administration.

In all the in vivo studies, treatments were administered by gastric intubation between 9 and 10 am in a volume between 1.5 and 2 mL by oral gavage.

All animal protocols followed in this study were approved by the Animal Ethics Review Committee for Animal Experimentation of the Universitat Rovira i Virgili and further approved by Generalitat de Catalunya (permission number 10780). 

### 2.6. Statistical Analysis

BP differences produced by the administration of the different WL were analyzed by a two-way analysis of variance (ANOVA) for the studies with SHR and WKY rats. Student’s T-test was used to evaluate differences between CWL from different campaigns in both IC_50_ analysis and antihypertensive study. A one-way ANOVA was used to evaluate differences between phenolic compounds in WL. All the analyses were performed using GraphPad Prism 7.04 for Windows (GraphPad Software, San Diego, California). Outliers were determined by using Grubbs’ test. Differences between groups were considered significant when *p* < 0.05. 

## 3. Results

### 3.1. Selection of the Wine Lees

Table 1 shows the ACEi activity of the five WL used in this study. As it is shown, WL obtained from red grape varieties (Cabernet, Garnacha, Mazuela, and Merlot) showed a greater ACEi activity than the one related to the white grape variety (Macabeo). Specifically, red grapes showed a percentage of ACEi activity between 28% and 56%. In addition, the concentration of WL needed to inhibit 50% of the ACE activity (IC_50_) was also determined. They ranged between 0.15 ± 0.01 and 3.74 ± 0.05 µL. Cabernet, Garnacha, and Mazuela WL were the ones with the highest activities (lower than 0.5 µL, Table 1) and were selected for further studies. The dose-response of ACE inhibition of some of the tested WL is represented in Figure 2.

### 3.2. Determination of the Phenolic Profile of the Three Selected Wine Lees

Phenolic composition of the selected WL was determined using commercial standards. As standards of all the phenolic compounds were not always available, a tentative quantification of these other compounds was carried out using the calibration curve of the most similar available structures. Figure 3 shows the results of the overlapped extract ion chromatograms (EIC) of non-anthocyanin phenolic compounds analyzed by UHPLC-(ESI-)-Q-TOF-MS (Figure 3A) and anthocyanin phenolic compounds analyzed by UHPLC-(ESI +)-Q-TOF-MS (Figure 3B). Table 2 shows the total phenolic content and total content of flavanols, flavonols, phenolic acids, stilbenes, and anthocyanins of Cabernet, Mazuela, and Garnacha WL. The total content of phenolic compounds in the CWL was almost double than the content measured in Mazuela and Garnacha WL (690.6, 395.3, and 379.6 mg/L, respectively). In addition, the contribution of the different phenolic families to the total phenolic content was different depending on the type of WL. Flavanols was the main family in the CWL distantly followed by anthocyanins and phenolic acids (311.1, 153.5, and 133.5 mg/L, respectively). However, flavanols and phenolic acids, in the same proportion, were the main groups in Mazuela and Garnacha WL. The main difference found between CWL and both Mazuela and Garnacha WL was the highest content of flavanols and anthocyanins showed by CWL (Table 2).

The sample individual phenolic profile on flavanols, flavonols, phenolic acids, and stilbenes is shown in Table 3. The major compounds in all samples were catechin, epicatechin, procyanidin dimer B2, and procyanidin dimer iso1, with higher levels found in CWL compared to Mazuela and Garnacha WL. Regarding the content of the other phenolic families, the major compounds found were: quercetin and isorhamnetin in the flavonols group, gallic acid in the phenolic acid group, and trans-resveratrol and piceatannol in the stilbene group. 

Regarding the anthocyanin composition (Table 4), a total of forty different anthocyanins were identified in the WL with malvidin-3-glucoside > malvidin-(6-acetyl)-3-glucoside > malvidin-(6-coumaroyl)-3-glucoside as the major compounds. The content of these three compounds was notably higher in the CWL compared to the other WL. 

### 3.3. Effect of Different Wine Lees on Blood Pressure in Hypertensive Rats

The antihypertensive effect of Cabernet, Garnacha, and Mazuela WL was evaluated in SHR rats after an acute oral dose (5 mL/kg bw). SBP and DBP results of this study are shown in Figure 4A,B, respectively. As expected, animals that received water did not show changes in their BP. In contrast, Captopril administration (50 mg/kg bw) led to a continuous decrease in the animals’ SBP and DBP 2 h post-treatment. The maximum decreases were observed at 6 h (−43.2 ± 3.9 and −47.2 ± 1.5 mmHg for SBP and DBP, respectively). Regarding the WL, only CWL showed an antihypertensive effect on both SBP and DBP in SHR, being their behavior similar to the one observed by Captopril. The maximum decrease in BP was also observed at 6 h post-administration (−36.4 ± 3.4 and −38.8 ± 4.6 mmHg for SBP and DBP, respectively). Initial BP values were recovered at 24 or 48 h for SBP and DBP, respectively. No significant changes in BP were found between the Garnacha or Mazuela WL groups and water group (Figure 4A,B). CWL were selected according their antihypertensive effect for further studies.

### 3.4. Effect of Cabernet Wine Lees on Blood Pressure in Normotensive Rats

The effect of CWL on BP was also evaluated in normotensive rats (WKY) in order to discard possible hypotensive effects. Initial values of SBP and DBP were 119.1 ± 4.2 and 87.9 ± 8.8 mmHg, respectively. The administration of a single dose of CWL (5 mL/kg bw) did not modify SBP or DBP values in the animals during the experiment (Figure 5). BP values were significantly similar to those showed by the animals that ingested water.

### 3.5. Variability between Cabernet Wine Lees from Two Different Harvests

Finally, the variability of ACEi and antihypertensive activities of CWL harvested in two different years (CWL and CWL2) were evaluated. ACEi activity (%) and IC_50_ did not show differences between CWL from different grape harvests (Figure 6A,B). The antihypertensive properties of CWL and CWL2 were also evaluated in SHR at a single dose of 5 mL/kg bw at 6 h post-administration. No differences were found in SBP and DBP between different harvests (Figure 6C,D).

## 4. Discussion

Annually, a large number of agri-food by-products are generated during food processing and their valorization has attracted a great deal of attention over the past few years [9]. In fact, one of the most emerging purposes is to be used as a source of bioactive compounds. These compounds are highly valued by the food, pharmaceutical, and cosmetic industries because they show a wide range of beneficial health effects. In this sense, winery by-products have been successfully used to obtain antioxidant [31], antimicrobial [32], anti-inflammatory [33], antihyperglycemic [34], or antihypertensive compounds [14,16,17,35]. For instance, extracts rich in phenolic compounds (proanthocyanidins or resveratrol), with antihypertensive properties in both rats and humans, have been extracted from stem, grape seeds, or skin, respectively [13,15,16,17,36,37,38]. However, to our knowledge, no studies have been performed to evaluate whether WL can present ACEi or antihypertensive properties. Thus, the aim of this study was to evaluate the ACEi and antihypertensive activities in several WL. For this, five WL samples were obtained in the elaboration of wine with a single grape variety (red grapes varieties: Cabernet, Garnacha, Mazuela, Merlot, and white grape variety: Macabeo) were selected to determine their ability to inhibit ACE. ACE inhibitors, as Captopril, Enalapril, or Lisinopril, are usually used to treat hypertension [39]. In fact, ACEi activity is commonly used as a screening tool in the search for natural antihypertensive compounds. The determination of the ACEi activity of the five WL showed that lees obtained from red grape varieties exerted higher activity than those obtained from the white grape variety (Table 1; Figure 2). Phenolic compounds are present in red grapes in larger quantities than in white grapes [40]. Therefore, these compounds could be responsible for the ACEi effects since in vitro studies have demonstrated inhibitory properties of phenolic compounds on ACE [8]. Similar results were reported by Pozo-Bayón et al. and Alcaide-Hidalgo et al., who studied the ACEi activity of red (Tempranillo) and white (Airén, Verdejo and Sauvignon Blanc) wines, respectively [41,42]. The ACEi activity of the WL was also calculated as IC_50_. The expression of this value in volume is indicative of the microliters of the WL necessary to inhibit the enzyme by 50% under the assay conditions, where the total volume is 240 μL. Therefore, it is a measure of the pharmacological potency, given that the lower the IC_50_, expressed in volume, the higher the potency of ACE inhibition of the WL assayed [43]. All WL obtained from red grapes showed a potent ACEi activity, although differences in the IC_50_ values can be noted (Table 1). ACEi activities of red WL ranged among 0.15 ± 0.0 and 0.32 ± 0.0 µL. These inhibitory potencies are higher than those reported by other authors in milk fermented with *Enterococcus*
*faecalis* and *Lactobacillus helveticus* [43,44]. These results clearly reveal important ACEi potency in WL obtained from red grapes. Although ACEi activity is frequently used to select antihypertensive compounds, it does not always correspond to an *in vivo* effect. Gastrointestinal digestion could produce variations over different WL compounds. In addition, first pass metabolism and microbiota metabolism will significantly modify the ingested phenolic compounds [45]. All these modifications could lead to changes in their ACEi properties. Therefore, *in vivo* studies must be carried out to demonstrate the antihypertensive effect of WL. In the present study, the antihypertensive effect of Cabernet, Garnacha, and Mazuela WL, selected by their ACEi activity, was evaluated in SHR after a single oral dose of 5 mL/kg bw (Figure 4). While Garnacha and Mazuela WL did not exhibit BP-lowering effects, CWL showed a clear antihypertensive effect. The maximum antihypertensive effect was reached at 6 h post-administration. The ACE inhibitor Captopril also exhibited a similar response time. A maximum decrease in BP at 4 or 6 h post-administration has been also reported by our group for other natural ACE inhibitors such as phenolic rich cocoa or grape seed extracts [30,46] or bioactive peptides [47]. Furthermore, the powerful antihypertensive effect of the CWL was similar to that observed for the drug Captopril. Phenolic extracts obtained from grape seeds (GSPE) have shown similar antihypertensive effects, with a maximum drop in BP at 6 h post-administration [15,17]. Similarly, Valls et al. observed a significant enhancement of endothelial function 5 h after administration to volunteers of a phenolic-enriched olive oil. In addition, this effect correlated with an increase of phenolic-derived metabolites in blood 2 h after its intake [48]. Notably, the BP-lowering effect produced by CWL (approximately 30 mmHg) could be a promising result since small reductions in BP may have an important impact on cardiovascular events in the hypertensive population [49]. In this sense, a reduction in 5 mmHg for DBP and 10 mmHg for SBP produces a significant reduction in the risk of suffering or worsening CVD [50,51]. The effect on BP of CWL was also tested in normotensive rats in order to rule out hypotensive effects. CWL did not modify the BP of these animals (Figure 5). This indicates that the antihypertensive effect of these CWL is specific to the hypertensive condition. 

In order to understand the different BP-lowering effects exhibited by the tested WL in rats, the phenolic profile of these three samples was studied. Phenolic compounds have been widely investigated due to their large number of beneficial properties, such as their cardioprotective effect [52], in which antihypertensive activity is included [53]. Specifically, a meta-analysis focused on grape phenolic compounds showed that their daily consumption reduced SBP by 1.48 mmHg when compared with the control group [54]. Thus, the different BP-lowering effects exhibited by the three WL would lie within the phenolic composition. According to this, results revealed that CWL contained twice the amount of total phenolic compounds when compared to Mazuela or Garnacha WL. These results highlight the importance of phenolic compounds for the antihypertensive activity of the WL. Specifically, it can be observed a higher concentration of flavanol and anthocyanin families in CWL compared to Mazuela or Garnacha WL (Table 2). Numerous studies have shown that the intake of flavanol-rich foods such as cocoa, red grapes, and red wine can be associated with improved vascular function and can repair and reduce BP in both hypertensive and pre-hypertensive individuals [55]. It has also been reported that flavanol-rich extracts from grape seed or cocoa showed antihypertensive effect after their acute [15,17,30,46] and chronic administration [16] to hypertensive rats. Furthermore, the flavanol monomers epicatechin and catechin have shown antihypertensive effect in both rats and humans administered at low concentrations [56,57,58]. The high levels of catechin, epicatechin, and procyanidins (Table 3) present in CWL suggest that these compounds could be in part responsible for the BP-lowering effect of this variety of WL. In addition, the polyphenol family of anthocyanins has also shown cardioprotective and antihypertensive properties [59,60]. Their circulating metabolites have also been directly related to vascular benefits [61]. Moreover, malvidin-3-glucoside, a compound from the anthocyanins family, has been reported as a potent vasodilator [62]. In our study, the anthocyanin content in CWL was higher than in the other WL. The main differences were in the levels of malvidin-3-glucoside, malvidin-(6-acetyl)-3-glucoside, and malvidin-(6-coumaroyl)-3-glucoside (Table 4). Therefore, anthocyanin family and specifically these compounds could be also involved in the BP-lowering effect of CWL. 

Finally, the ACEi and antihypertensive activities of a CWL coming from a different harvest were also studied. No significant differences in the ACEi activity or in the decrease of BP produced by the different CWL were observed (Figure 6), indicating good reproducibility of the CWL beneficial effects. 

## 5. Conclusions

This study shows that WL from red grapes present a potent ACEi activity and that CWL, WL coming from the grape variety Cabernet, also exhibited a potent BP-lowering effect, specific to a hypertensive condition. It is noteworthy that in this study CWL were administered at 5 mL/kg bw. This dose corresponds to an intake of 73 mL/day in humans, using a translation of animal to human doses [63] and estimating the daily intake for an adult human with body weight 70 kg and body height 175 cm. Although experimental results obtained in animals cannot be directly translatable to humans, the fact that only 73 mL of CWL exhibit antihypertensive effects opens the door to the valorization of CWL by their BP-lowering properties. Nevertheless, the quantity of CWL necessary to decrease arterial BP in humans should be definitively established when clinical trials are conducted. CWL antihypertensive activity has been related to their highest content in anthocyanins and flavanols. In addition, these beneficial effects were reproducible in CWL from different vintages. These findings open the door to the use of CWL to alleviate hypertension. At the same time, this study would also allow the wine industry to revalue by-products as WL and, therefore, reduce their associated environmental problems. However, HTN is a chronic pathology that requires chronic treatment; thus, chronic studies are necessary to evaluate of antihypertensive effect of long-term administration of CWL.

## 6. Patents

Patent application “Wine lees, derivatives thereof and their uses”: application number EP20382358.8 and PCT/EP2021/053051.

## Figures and Tables

**Figure 1 nutrients-13-00679-f001:**
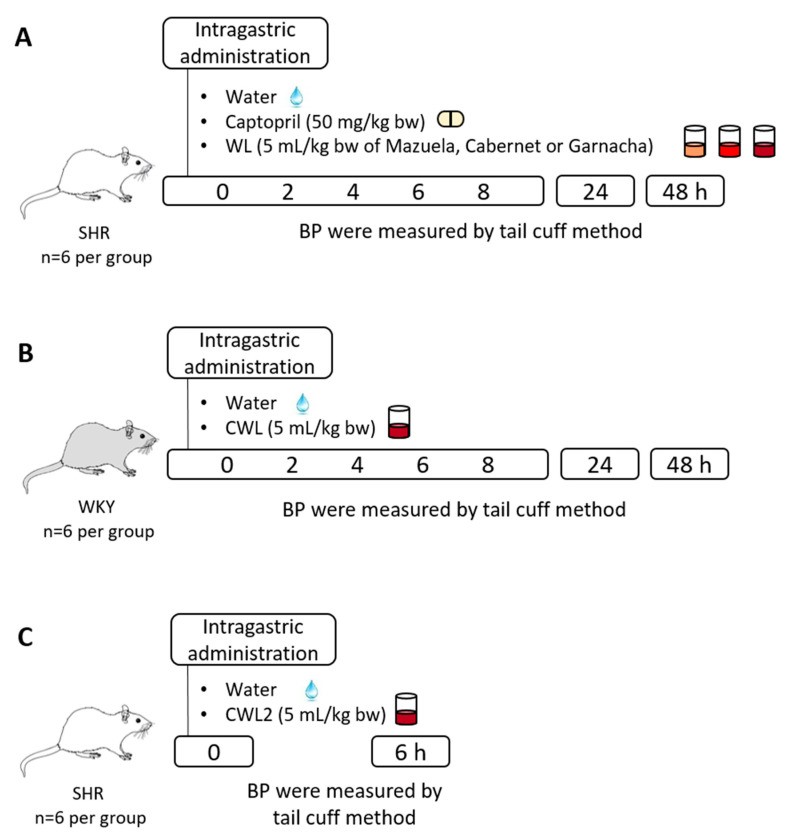
Graphical representation of the three in vivo studies carried out to investigate (**A**) the effect of three wine lees (WL) on blood pressure (BP) in spontaneously hypertensive rats (SHR), (**B**) the effect of Cabernet WL (CWL) on BP in normotensive Wistar–Kyoto rats (WKY) and (**C**) the effect of CWL from a different harvest (CWL2) at 6 h post-administration in SHR.

**Figure 2 nutrients-13-00679-f002:**
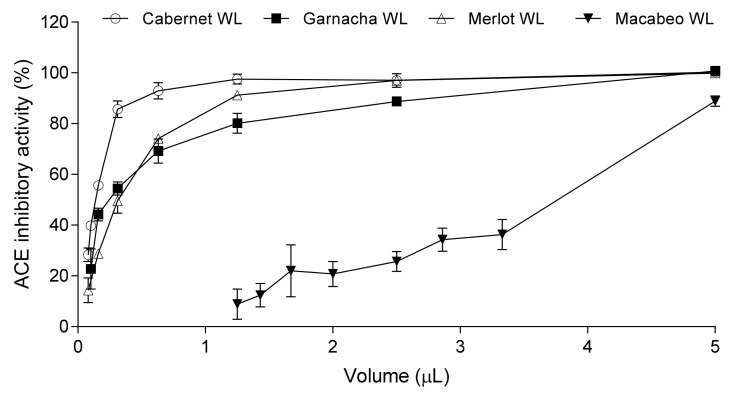
Dose–response curves (effect as a function of the dose in µL) for Cabernet, Garnacha, Merlot, and Macabeo wine lees (WL). Values are the average of three replicates ± SD.

**Figure 3 nutrients-13-00679-f003:**
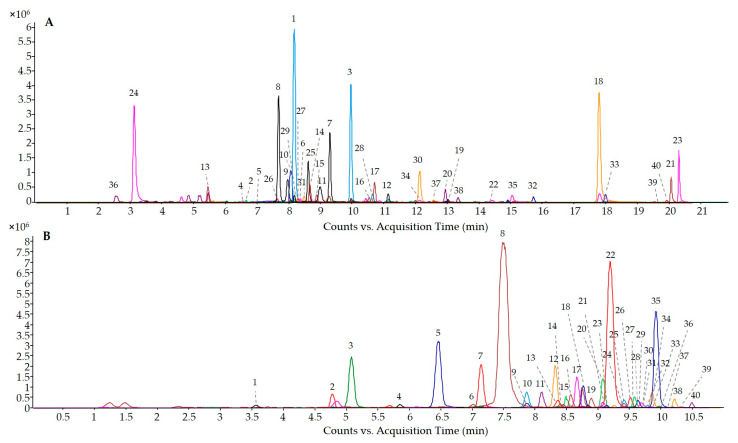
Overlapped extract ion chromatograms (EIC) of (**A**) non-anthocyanin wine lees (WL)-phenolic compounds analyzed by UHPLC-(ESI-)-Q-TOF-MS and (**B**) anthocyanin WL-phenolic compounds analyzed by UHPLC-(ESI + )-Q-TOF-MS. Chromatographic peaks are numbered according to Table 3 and Table 4.

**Figure 4 nutrients-13-00679-f004:**
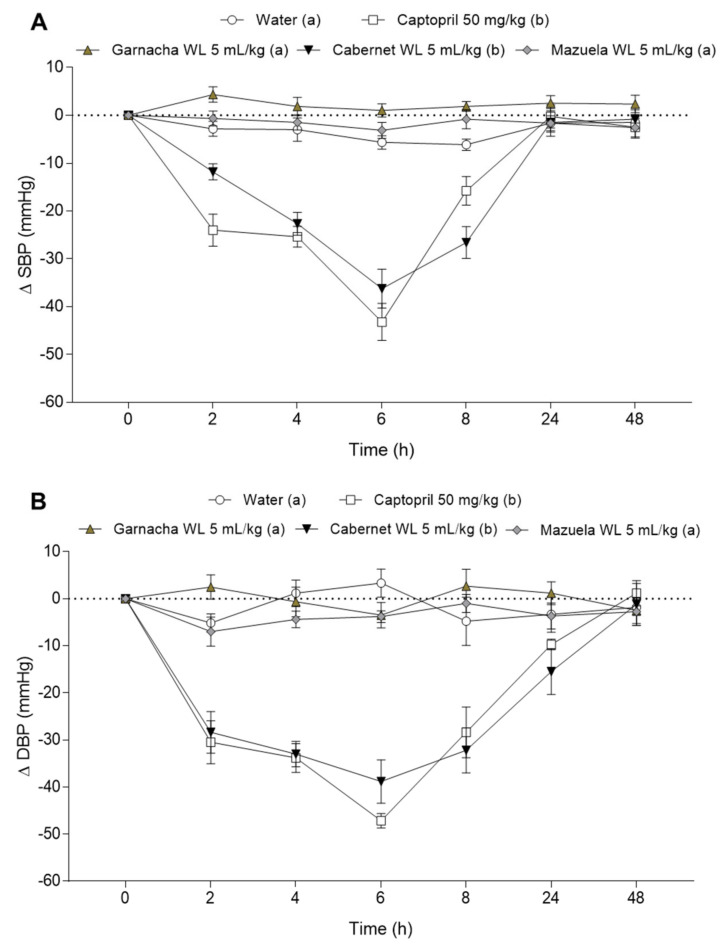
Decrease of systolic blood pressure (SBP, **A**) and diastolic blood pressure (DBP, **B**) in spontaneous hypertensive rats after the administration of water, Captopril (50 mg/kg bw), and the three selected wine lees (WL; 5 mL/kg bw): Garnacha WL, Cabernet WL, and Mazuela WL. Data are expressed as mean (*n* = 6) ± SEM. Significant differences (*p* < 0.05) between treatments are represented by different letters in the legend. *p* value was estimated by two-way ANOVA and Tukey test was used as post hoc.

**Figure 5 nutrients-13-00679-f005:**
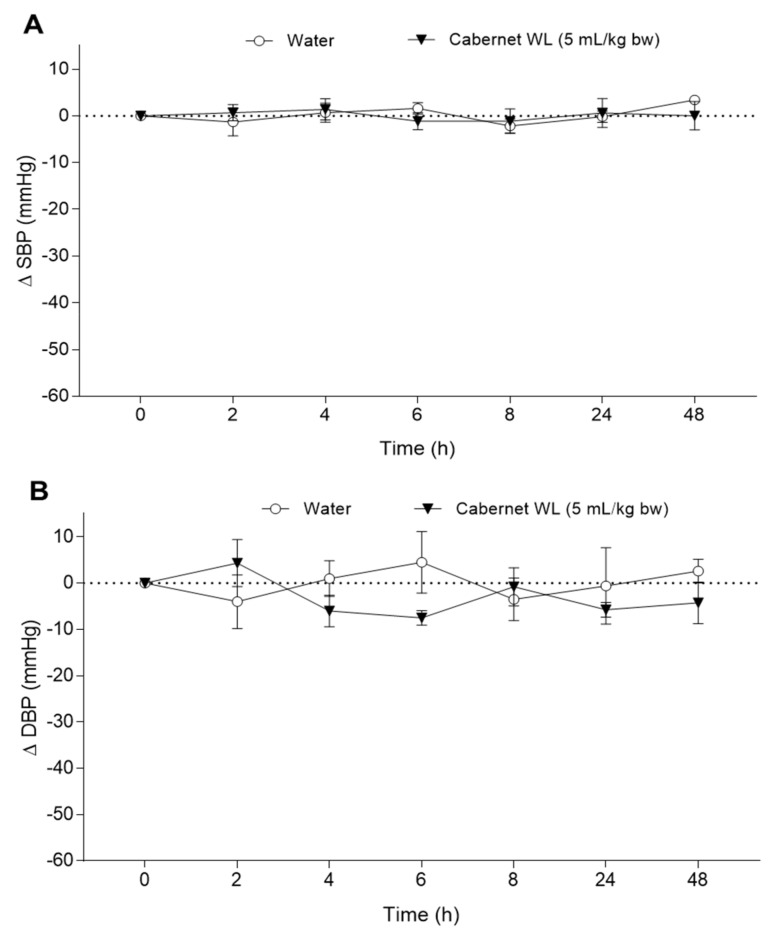
Decrease in systolic blood pressure (SBP, **A**) and diastolic blood pressure (DBP, **B**) caused in Wistar–Kyoto rats after the acute administration of water or Cabernet WL (5 mL/g bw). Data are expressed as mean (*n* = 6) ± SEM. No significant differences (*p* < 0.05) were found. *p* was estimated by two-way ANOVA.

**Figure 6 nutrients-13-00679-f006:**
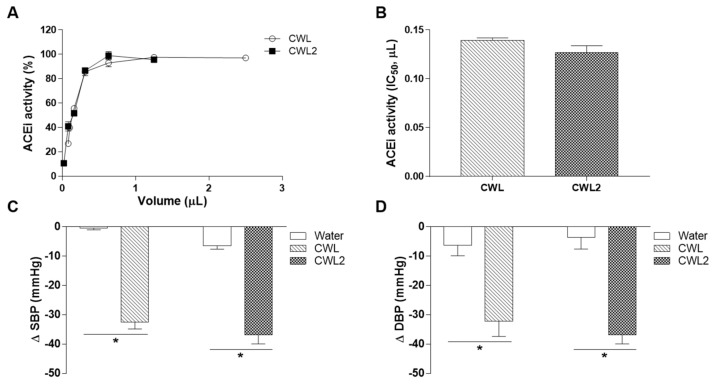
Variability between Cabernet wine lees (CWL) from two different harvests (CWL and CWL2). ACEi activity for both harvests is represented as dose–response curve (%, **A**) and IC_50_ (**B**). Data are shown as mean ± SD (*n* = 3). No significant differences (*p* < 0.05) were found between IC_50_ values (Student’s T-test). Decrease in systolic blood pressure (SBP, **C**) and diastolic blood pressure (DBP, **D**) caused in spontaneous hypertensive rats by the acute administration of water, CWL, or CWL2 (both 5 mL/kg bw). Data are shown as mean ± SEM (*n* = 6). No significant differences (*p* < 0.05) were found between CWL and CWL2 (Student’s T-test). * represents significant differences (*p* < 0.05) between CWL and CWL2 and their respective water control groups estimated by Student’s T-test.

**Table 1 nutrients-13-00679-t001:** Angiotensin-converting enzyme inhibitory (ACEi) activity of the wine lees obtained in the winemaking process using different individual grape varieties.

Grape Variety	ACEi activity	
	%*	IC_50_ (µL)
Cabernet	55.69 ± 1.92	0.15 ± 0.01
Garnacha	44.16 ± 2.54	0.22 ± 0.01
Mazuela	50.70 ± 8.10	0.21 ± 0.03
Merlot	28.76 ± 0.34	0.32 ± 0.02
Macabeo	< 10	3.74 ± 0.05

*ACEi activity showed by a wine lees volume of 0.16 µL.

**Table 2 nutrients-13-00679-t002:** Total composition of flavanols, flavonols, phenolic acids, stilbenes, and anthocyanins in studied wine lees (WL).

Phenolic Compounds	Cabernet WL (mg/L)	Mazuela WL (mg/L)	Garnacha WL (mg/L)
Flavanols	331.11	122.63	154.60
Flavonols	57.62	44.83	57.28
Phenolic acids	133.54	132.75	103.05
Stilbenes	14.73	20.08	13.59
Anthocyanins	153.53	74.99	51.12
Total	690.63	395.28	379.64

**Table 3 nutrients-13-00679-t003:** Characterization of phenolic compounds in Cabernet, Mazuela, and Garnacha wine lees (WL) by UHPLC-(ESI-)-Q-TOF-MS.

	Compounds	R.T. (min)	[M-H]-	Fragment (*m/z*)	Cabernet WL (mg/L)	Mazuela WL (mg/L)	Garnacha WL (mg/L)
	Flavanols						
**1**	Catechin	8.17	289.0718		97.63 ± 0.62 ^a^	39.27 ± 0.25 ^b^	56.65 ± 0.36 ^c^
**2**	Catechin gallate ^1^	6.66	441.0827	289.07209	0.80 ± 0.01 ^a^	1.53 ± 0.03 ^b^	0.79 ± 0.01 ^a^
**3**	Epicatechin	9.96	289.0718		43.48 ± 0.23 ^a^	12.94 ± 0.07 ^b^	20.16 ± 0.11 ^b^
**4**	(Epi)catechin O-glucoside iso1 ^2^	6.55	451.1246	289.0721	0.50 ± 0.00 ^a^	0.50 ± 0.00 ^a^	0.90 ± 0.00 ^a^
**5**	(Epi)catechin O-glucoside iso2 ^2^	7.41	451.1246	289.0721	0.33 ± 0.00 ^a^	0.29 ± 0.00 ^a^	0.43 ± 0.00 ^a^
**6**	(Epi)catechin O-glucoside iso3 ^2^	8.37	451.1246	289.0721	1.47 ± 0.03 ^a^	0.77 ± 0.01 ^b^	1.64 ± 0.03 ^a^
**7**	Procyanidin dimer B2	9.30	577.1387	289.0733	34.60 ± 0.01 ^a^	9.61 ± 0.00 ^b^	9.09 ± 0.00 ^c^
**8**	Procyanidin dimer iso1 ^3^	7.68	577.1387	289.0733	64.21 ± 0.34 ^a^	32.11 ± 0.17 ^b^	32.37 ± 0.17 ^b^
**9**	Procyanidin dimer iso2 ^3^	7.97	577.1387	289.0733	14.26 ± 0.09 ^a^	4.19 ± 0.03 ^b^	5.81 ± 0.04 ^c^
**10**	Procyanidin dimer iso3 ^3^	8.18	577.1387	289.0733	2.96 ± 0.02 ^a^	0.84 ± 0.01 ^b^	1.50 ± 0.01 ^c^
**11**	Procyanidin dimer iso4 ^3^	8.99	577.1387	289.0733	12.80 ± 0.00 ^a^	2.71 ± 0.00 ^b^	3.79 ± 0.00 ^c^
**12**	Procyanidin dimer iso5 ^3^	11.14	577.1387	289.0733	4.32 ± 0.04 ^a^	1.54 ± 0.02 ^b^	1.97 ± 0.02 ^b^
**13**	Procyanidin trimer iso1 ^3^	5.46	865.2016	577.1369	16.28 ± 0.29 ^a^	5.14 ± 0.09 ^b^	7.91 ± 0.14 ^c^
**14**	Procyanidin trimer iso2 ^3^	8.67	865.2016	577.1369	14.35 ± 0.71 ^a^	4.62 ± 0.23 ^b^	5.15 ± 0.25 ^c^
**15**	Procyanidin trimer iso3 ^3^	8.89	865.2016	577.1369	6.11 ± 0.03 ^a^	2.37 ± 0.01 ^b^	2.55 ± 0.01 ^b^
**16**	Procyanidin trimer iso4 ^3^	10.55	865.2016	577.1369	3.22 ± 0.14 ^a^	1.38 ± 0.06 ^b^	1.19 ± 0.06 ^b^
**17**	Procyanidin trimer iso5 ^3^	10.71	865.2016	577.1369	13.79 ± 0.23 ^a^	2.82 ± 0.05 ^b^	2.70 ± 0.05 ^b^
	Flavonols						
**18**	Quercetin	17.80	301.0372		36.78 ± 0.17 ^a^	28.62 ± 0.13 ^b^	25.35 ± 0.12 ^c^
**19**	Quercetin-3-O-glucoside ^4^	13.00	463.0904	301.0361	1.63 ± 0.04 ^a^	2.73 ± 0.08 ^b^	7.68 ± 0.21 ^c^
**20**	Quercetin-3-O-glucuronide ^4^	12.95	477.0702	301.0369	2.42 ± 0.01 ^a^	5.62 ± 0.03 ^b^	4.86 ± 0.02 ^c^
**21**	Kaempferol ^4^	20.07	285.0405		5.15 ± 0.02 ^a^	1.63 ± 0.01 ^b^	9.35 ± 0.04 ^c^
**22**	kaempferol-3-O-glucuronide ^4^	14.22	461.0763	285.0412	0.48 ± 0.01 ^a^	1.31 ± 0.02 ^b^	3.24 ± 0.05 ^c^
**23**	Isorhamnetin ^4^	20.31	315.0531		11.16 ± 0.12 ^a^	4.92 ± 0.05 ^b^	6.80 ± 0.07 ^c^
	Phenolic acids						
**24**	Gallic acid	3.13	169.0193		120.87 ± 3.67 ^a^	121.19 ± 3.68 ^a^	96.11 ± 2.92 ^b^
**25**	Caffeic acid	8.63	179.0401		3.27 ± 0.04 ^a^	2.18 ± 0.02 ^a^	1.09 ± 0.01 ^a^
**26**	Caffeic acid O-glucoside iso1 ^5^	7.64	341.0878	179.0350	0.55 ± 0.02 ^a^	1.21 ± 0.05 ^a^	0.13 ± 0.01 ^a^
**27**	Caffeic acid O-glucoside iso2 ^5^	8.29	341.0878	179.0350	0.66 ± 0.03 ^a^	1.13 ± 0.05 ^a^	0.20 ± 0.01 ^a^
**28**	p-Coumaric acid	10.65	163.0439		3.44 ± 0.03 ^a^	3.73 ± 0.04 ^a^	1.15 ± 0.01 ^a^
**29**	4-Hydroxybenzoic acid	8.17	137.0243		1.67 ± 0.05 ^a^	0.89 ± 0.03 ^a^	1.19 ± 0.04 ^a^
**30**	Ferulic acid	12.00	193.0506		0.75 ± 0.01 ^a^	0.27 ± 0.00 ^a^	0.51 ± 0.01 ^a^
**31**	Vanillic acid	8.51	167.0350		2.33 ± 0.07 ^a^	2.15 ± 0.08 ^a^	2.67 ± 0.04 ^a^
	Stilbenes						
**32**	trans-Resveratrol ^6^	15.73	227.0714		4.60 ± 0.02 ^a^	3.63 ± 0.02 ^b^	3.12 ± 0.01 ^c^
**33**	Resveratrol iso1^6^	18.00	227.0714		2.95 ± 0.01 ^a^	2.56 ± 0.01 ^b^	0.97 ± 0.00 ^c^
**34**	Resveratrol O-glucoside iso1 ^6^	12.44	389.1242	227.0721	0.27 ± 0.00 ^a^	1.22 ± 0.02 ^b^	1.17 ± 0.02 ^b^
**35**	Resveratrol O-glucoside iso2 ^6^	14.92	389.1242	227.0721	1.35 ± 0.02 ^a^	6.01 ± 0.10 ^b^	3.32 ± 0.05 ^c^
**36**	Piceatannol ^6^	2.59	243.0663	203.0727	4.20 ± 0.05 ^a^	4.96 ± 0.06 ^b^	4.04 ± 0.05 ^c^
**37**	Piceatannol 3-O-glucoside iso1 ^6^	12.89	405.1208	243.0670	0.22 ± 0.01 ^a^	0.14 ± 0.00 ^b^	0.04 ± 0.00 ^c^
**38**	Piceatannol 3-O-glucoside iso2 ^6^	13.15	405.1208	243.0670	0.06 ± 0.00 ^a^	0.18 ± 0.00 ^b^	0.05 ± 0.00 ^a^
**39**	Viniferin-iso1 ^6^	19.53	453.1344	116.9291	0.27 ± 0.01 ^a^	0.33 ± 0.01 ^a^	0.15 ± 0.00 ^b^
**40**	Viniferin-iso2 ^6^	19.92	453.1344	116.9291	0.81 ± 0.02 ^a^	1.05 ± 0.03 ^b^	0.73 ± 0.02 ^c^

Abbreviations: Retention time (R.T.). Wine lees (WL). ^a,b,c^ Different letters indicate significant differences in the content of each individual phenolic compound between the different WL (*p* < 0.05; one-way ANOVA).^1^ Tentatively quantified using the catechin calibrating curve. ^2^ Tentatively quantified using the epicatechin calibrating curve. ^3^ Tentatively quantified using the procyanidin dimer B2 calibrating curve. ^4^ Tentatively quantified using the quercetin calibrating curve. ^5^ Tentatively quantified using the caffeic acid calibrating curve. ^6^ Tentatively quantified using the resveratrol calibrating curve.

**Table 4 nutrients-13-00679-t004:** Characterization of anthocyanin in Cabernet, Mazuela, and Garnacha wine lees (WL) by UHPLC- (ESI +)-Q-TOF-MS.

	Anthocyanins	R.T. (min)	[M-H]+	Fragment (*m/z*)	Cabernet WL (mg/L)	Mazuela WL (mg/L)	Garnacha WL (mg/L)
**1**	Gallocatechin-Malvidin-3-glucoside dimer ^1^	3.58	797.2035		0.25 ± 0.01 ^a^	0.10 ± 0.00 ^a^	0.16 ± 0.00 ^a^
**2**	Malvidin-3-glucoside-(epi) catechin ^1^	4.84	781.1974		1.11 ± 0.01 ^a^	0.53 ± 0.00 ^b^	0.50 ± 0.00 ^b^
**3**	Delphinidin-3-glucoside ^2^	5.06	465.1028	303.0511	3.69 ± 0.04 ^a^	2.98 ± 0.03 ^b^	1.39 ± 0.01 ^c^
**4**	Cyanidin-3-glucoside ^2^	5.85	449.1078	287.0531	0.23 ± 0.01 ^a^	0.20 ± 0.01 ^a^	0.16 ± 0.01 ^a^
**5**	Petunidin-3-glucoside ^3^	6.47	479.1184	317.0669	5.03 ± 0.06 ^a^	4.90 ± 0.06 ^a^	2.18 ± 0.03 ^b^
**6**	Petunidin-3-glucoside-pyruvic acid ^3^	7.05	547.1082	385.0547	0.09 ± 0.00 ^a^	0.06 ± 0.00 ^a^	0.03 ± 0.00 ^a^
**7**	Peonidin-3-glucoside ^3^	7.14	463.1235	301.0717	2.72 ± 0.04 ^a^	1.83 ± 0.03 ^b^	2.48 ± 0.04 ^c^
**8**	Malvidin-3-glucoside ^1^	7.48	493.1341	331.0843	60.67 ± 0.68 ^a^	43.90 ± 0.49 ^b^	26.78 ± 0.30 ^c^
**9**	Peonidin-3-glucoside-pyruvic acid ^3^	7.81	531.1133	369.0607	0.04 ± 0.00 ^a^	0.02 ± 0.00 ^a^	0.02 ± 0.00 ^a^
**10**	Delphinidin-(6-acetyl)-3-glucoside ^2^	7.87	507.1133	303.0496	0.91 ± 0.02 ^a^	0.09 ± 0.00 ^b^	0.02 ± 0.00 ^b^
**11**	Visitin A (malvidin-3-glucoside-pyruvic acid) ^1^	8.11	561.1239	399.0730	1.23 ± 0.01 ^a^	0.63 ± 0.01 ^b^	0.35 ± 0.00 ^c^
**12**	Visitin B (malvidin-3-glucoside-acetaldehyde) ^1^	8.32	517.1341	355.0826	3.06 ± 0.08 ^a^	4.92 ± 0.12 ^b^	5.11 ± 0.00 ^b^
**13**	Malvidin-3-glucoside-ethyl-(epi) catechin ^1^	8.40	809.2287		0.37 ± 0.00 ^a^	0.09 ± 0.00 ^b^	0.31 ± 0.13 ^a^
**14**	Cyanidin-(6-acetyl)-3-glucoside ^2^	8.45	491.1184	491.1189	0.20 ± 0.00 ^a^	0.02 ± 0.00 ^b^	0.01 ± 0.00 ^b^
**15**	Acetylvisitin A ^1^	8.50	603.1344	399.0718	0.79 ± 0.03 ^a^	0.10 ± 0.00 ^b^	0.15 ± 0.00 ^b^
**16**	Malvidin-3-glucoside-ethyl-(epi) catechin ^1^	8.57	809.2287		1.38 ± 0.02 ^a^	0.51 ± 0.01 ^b^	1.65 ± 0.00 ^c^
**17**	Petunidin-(6-acetyl)-3-glucoside ^3^	8.66	521.1378	317.0667	1.29 ± 0.04 ^a^	0.16 ± 0.01 ^b^	0.04 ± 0.02 ^b^
**18**	Malvidin-3-glucoside-ethyl-(epi) catechin ^1^	8.75	809.2287		2.04 ± 0.06 ^a^	0.80 ± 0.03 ^b^	2.63 ± 0.00 ^c^
**19**	Acetylvisitin B ^1^	8.77	559.1446	355.0813	1.66 ± 0.05 ^a^	0.47 ± 0.01 ^b^	0.27 ± 0.08 ^b^
**20**	Peonidin-(6-acetyl)-3-glucoside ^3^	9.08	505.1341	301.0714	1.32 ± 0.03 ^a^	0.13 ± 0.00 ^b^	0.08 ± 0.01 ^b^
**21**	Delphinidin-(6-coumaroyl)-3-glucoside ^2^	9.08	611.1395	303.0508	0.44 ± 0.01 ^a^	0.55 ± 0.01 ^a^	0.09 ± 0.00 ^b^
**22**	Malvidin-(6-acetyl)-3-glucoside ^1^	9.13	535.1446	331.0836	28.39 ± 0.03 ^a^	2.57 ± 0.00 ^b^	0.79 ± 0.00 ^c^
**23**	Coumaroylvisitin A ^1^	9.29	707.1607	399.0718	0.20 ± 0.00 ^a^	0.13 ± 0.00 ^b^	0.04 ± 0.00 ^b^
**24**	Malvidin-(6-caffeoyl)-3-glucoside ^1^	9.41	655.1657	331.0808	0.36 ± 0.02 ^a^	0.10 ± 0.00 ^b^	0.04 ± 0.00 ^b^
**25**	Cyanidin-(6-coumaroyl)-3-glucoside ^2^	9.42	595.1446	287.0560	0.10 ± 0.00 ^a^	0.11 ± 0.00 ^a^	0.03 ± 0.00 ^b^
**26**	Catechin-ethyl-Malvidin-3-acetylglucoside dimer ^1^	9.43	851.2511		0.88 ± 0.03 ^a^	0.03 ± 0.00 ^b^	0.06 ± 0.00 ^b^
**27**	Petunidin-(6-coumaroyl)-3-glucoside ^3^	9.52	625.1552	317.0662	0.74 ± 0.03 ^a^	0.78 ± 0.01 ^a^	0.16 ± 0.00 ^b^
**28**	Pinotin A (malvidin-3-glucoside-vinylcatechol) ^1^	9.53	625.1552	463.0998	0.84 ± 0.02 ^a^	0.88 ± 0.02 ^a^	0.18 ± 0.00 ^b^
**29**	Malvidin-glucoside-vinyl-catechin ^1^	9.56	805.1974		0.15 ± 0.00 ^a^	0.08 ± 0.00 ^b^	0.16 ± 0.00 ^a^
**30**	Coumaroylvisitin B ^1^	9.58	663.1708	355.0822	0.91 ± 0.03 ^a^	1.08 ± 0.04 ^b^	1.12 ± 0.04 ^b^
**31**	Malvidin-3-glucoside-vinylguaiacol ^1^	9.63	639.1708	331.0823	0.59 ± 0.01 ^a^	0.37 ± 0.01 ^b^	0.17 ± 0.00 ^b^
**32**	Catechin-ethyl-malvidin-3-coumaroylglucoside dimer ^1^	9.70	955.2785		0.68 ± 0.01 ^a^	0.21 ± 0.00 ^b^	0.51 ± 0.01 ^a^
**33**	Catechin-ethyl-malvidin-3-acetylglucoside dimer ^1^	9.81	851.2511		0.14 ± 0.00 ^a^	0.02 ± 0.00 ^b^	0.02 ± 0.00 ^b^
**34**	Peonidin-(6coumaroyl)-3-glucoside ^3^	9.87	609.1603	301.0716	0.94 ± 0.03 ^a^	0.60 ± 0.02 ^b^	0.42 ± 0.01 ^c^
**35**	Malvidin-(6-coumaroyl)-3-glucoside ^1^	9.92	639.1708	331.0823	10.77 ± 0.02 ^a^	4.43 ± 0.01 ^b^	2.31 ± 0.01 ^c^
**36**	Malvidin-glucoside-vinyl-catechin ^1^	9.99	805.1974		0.16 ± 0.00 ^a^	0.06 ± 0.00 ^b^	0.14 ± 0.00 ^a^
**37**	Acetyl-pinotin A ^1^	10.19	667.1657		0.01 ± 0.00 ^a^	0.00 ± 0.00 ^b^	0.01 ± 0.00 ^a^
**38**	Malvidin 3-O-glucoside 4-vinylphenol (Pigment A) ^1^	10.22	609.1603	447.1079	0.64 ± 0.01 ^a^	0.44 ± 0.00 ^b^	0.44 ± 0.00 ^b^
**39**	Catechin-ethyl-malvidin-3-coumaroylglucoside dimer ^1^	10.33	955.2785		0.12 ± 0.00 ^a^	0.04 ± 0.00 ^b^	0.10 ± 0.00 ^a^
**40**	Malvidin acetyl 3-O-glucoside 4-vinylphenol (Acetyl-pigment A) ^1^	10.50	651.1708	447.1076	0.38 ± 0.01 ^a^	0.03 ± 0.00 ^b^	0.02 ± 0.00 ^b^

Abbreviations: Retention time (R.T.). Wine lees (WL). ^a,b,c^ Different letters indicate significant differences in the content of each individual phenolic compound between the different WL (*p* < 0.05; one-way ANOVA). ^1^ Tentatively quantified using the calibrating curve of malvidin glucoside. ^2^ Tentatively quantified using the calibrating curve of cyaniding rutinoside. ^3^ Tentatively quantified using the calibrating curve of peonidin rutinoside.

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
