# Peer review of "ACE Inhibitory and Antihypertensive Activities of Wine Lees and Relationship among Bioactivity and Phenolic Profile"

_nutrients, 2021, doi:10.3390/nu13020679_

Round 1
Reviewer 1 Report
This study examined the potential blood pressure lowering effects of wine lees intake in hypertensive rat model. Sustainable uses for agricultural waste that accumulate during food processing, such that occurs with winemaking, is currently of considerable interest. The current study reports rather impressive blood pressure lowering effects of wine lees from cabernet sauvignon. Apart from needed improvements in grammar, some comments follow:
- Please check for the aberrant insertions of hyphens throughout the text. A good example occurs on line 19 in the abstract where effect is hyphenated.
- Lines 176 through 190 can be removed, but you do need to report your ethical approval.
- More information on the animals and experimental design is needed. Please report how long they were housed before initialization of the experiments. Were they fasted? What was the average BP upon initiation of the experiments?
- An n=6 per experiment is reported. Were all the experiments conducted in the same 6 animals? If so, how long between experiments?
- Line 227 – please correct the spelling of anthocyanidins
- Statistical analysis & Figure 4– a multiple comparison post hoc test should also be employed. In Figure 4 it states that letters are used to state significant differences, but they are not there. Please report any significant changes over time along with between group differences.
- Table 4 is cut off and not showing the complete dataset.
- Figure 7 panels C and D: what is important here is that you do or do not have a significant difference between the two CWL. Please include the statistical results for between the two CWLs.
- Line 347 and 348: It is not gastrointestinal digestion that is the issue, but that first pass metabolism significantly and rapidly alters the parent compounds, which is why you cannot readily translate in vitro phenolic/polyphenol studies to the in vivo condition. Somewhere in the discussion you should note that the ACEi activity needs to be confirmed in vivo.
- Discussion – some discussion as to how the flavanol/anthocyanin intake in the current study relates to flavanol/anthocyanin intake by humans is needed. Note that the levels provided to the rats need to be metabolically scaled.
- Some discussion with regards to why maximal effects were observed at 6 hours is needed. Is this the typical maximal response time for ACE inhibition? And for the phenolic/polyphenols what forms are typically present in the circulation at this time after intake?
Author Response
Dear Reviewer 1,
Please find the point-by-point response to each of your comments with the description of the changes made in the manuscript.
Thank you very much for your attention reconsidering our manuscript for its publication in Nutrients.
Yours sincerely, Francisca I. Bravo

Reviewer 2 Report
comments in the attached word file.

Author Response
Dear Reviewer 2,
Please find the point-by-point response to each of your comments with the description of the changes made in the manuscript.
Thank you very much for your attention reconsidering our manuscript for its publication in Nutrients.
Yours sincerely, Francisca I. Bravo

Reviewer 3 Report
- This study was interesting, but limited evidence from animal model. Basically, much more information could be obtained from animal study, including angiotensin 1, angiotensin 2, ACE activity, and aldosterone. ACEi was measured by in vitro is not so suitable and can not response real situation.
- The statistical expression was not clearly in the figure.
- Please run the correlated coefficient analysis to figure out the potential contributor in WL
Author Response
Dear Reviewer 3,
Please find the point-by-point response to each of your comments with the description of the changes made in the manuscript.
Thank you very much for your attention reconsidering our manuscript for its publication in Nutrients.
Yours sincerely, Francisca I. Bravo

Round 2
Reviewer 3 Report
The revised one is well modified, and it is accepted.